# Multi-Class Cancer Subtyping in Salivary Gland Carcinomas with MALDI Imaging and Deep Learning

**DOI:** 10.3390/cancers14174342

**Published:** 2022-09-05

**Authors:** David Pertzborn, Christoph Arolt, Günther Ernst, Oliver J. Lechtenfeld, Jan Kaesler, Daniela Pelzel, Orlando Guntinas-Lichius, Ferdinand von Eggeling, Franziska Hoffmann

**Affiliations:** 1Innovative Biophotonics & MALDI Imaging, ENT Department, Jena University Hospital, 07747 Jena, Germany; 2Institute of Pathology, Medical Faculty, University of Cologne, 50937 Cologne, Germany; 3Department of Analytical Chemistry, Research Group BioGeoOmics, Helmholtz Centre for Environmental Research—UFZ, 04318 Leipzig, Germany; 4ProVIS−Centre for Chemical Microscopy, Helmholtz Centre for Environmental Research—UFZ, 04318 Leipzig, Germany

**Keywords:** MALDI imaging, deep learning, salivary gland carcinomas, explainable artificial intelligence

## Abstract

**Simple Summary:**

The correct diagnosis of different salivary gland carcinomas is important for a prognosis. This diagnosis is imprecise if it is based only on clinical symptoms and histological methods. Mass spectrometry imaging can provide information about the molecular composition of sample tissues. Using a deep-learning method, we analyzed the mass spectrometry imaging data of 25 patients. Using this workflow we could accurately predict the tumor type in each patient sample.

**Abstract:**

Salivary gland carcinomas (SGC) are a heterogeneous group of tumors. The prognosis varies strongly according to its type, and even the distinction between benign and malign tumor is challenging. Adenoid cystic carcinoma (AdCy) is one subgroup of SGCs that is prone to late metastasis. This makes accurate tumor subtyping an important task. Matrix-assisted laser desorption/ionization (MALDI) imaging is a label-free technique capable of providing spatially resolved information about the abundance of biomolecules according to their mass-to-charge ratio. We analyzed tissue micro arrays (TMAs) of 25 patients (including six different SGC subtypes and a healthy control group of six patients) with high mass resolution MALDI imaging using a 12-Tesla magnetic resonance mass spectrometer. The high mass resolution allowed us to accurately detect single masses, with strong contributions to each class prediction. To address the added complexity created by the high mass resolution and multiple classes, we propose a deep-learning model. We showed that our deep-learning model provides a per-class classification accuracy of greater than 80% with little preprocessing. Based on this classification, we employed methods of explainable artificial intelligence (AI) to gain further insights into the spectrometric features of AdCys.

## 1. Introduction 

Adenoid cystic carcinomas (AdCys) are a rare and heterogenous subgroup of head and neck carcinomas. The combination of rare occurrence and heterogeneity even between two samples of the same tumor subtype makes histological differentiation from other types of head and neck cancers challenging [1].

Previous studies have shown different levels of protein expressions in different types of salivary gland carcinomas (SGC) and MALDI imaging has successfully been used to confirm those results [2,3].

As a label-free imaging technique, MALDI imaging allows for the spatially resolved measurement of biomolecules in thin tissue sections. It has been demonstrated to be a useful tool for cancer research for the classification of different tumor species and the discovery of potential tumor marker proteins [4,5]

In combination with modern data analysis pipelines, MALDI imaging can be used for binary tumor classification as shown in [6,7,8]. Recent results attained with more advanced machine learning show the applicability of MALDI imaging for the histologically challenging subtyping of different epithelial ovarian cancers [9].

One of the most successful modern machine learning approaches is deep learning [10]. Deep learning describes a class of artificial neural networks with multiple hidden layers that are used to solve a multitude of problems ranging from scene recognition for self-driving cars to medical applications such as tumor classification and segmentation in magnetic resonance imaging [11,12,13]. A detailed overview of deep learning in medical imaging is given in [14]. The type of artificial neural networks most commonly used in medical image and signal processing tasks is convolutional neural networks (CNNs) [15]. With the increasing amount of labelled data available, this type of network has been used to achieve state-of-the-art performance in numerous tasks, such as image classification, speech recognition, and scene labeling [16]. In some medical tasks, such as the detection of breast cancer metastasis, CNN-based machine learning approaches already outperform human experts [17], [18] and show promising results in the classification of skin diseases [19].

With deep neural networks being used for critical decision-making in fields such as the aforementioned self-driving vehicles and tumor detection, there is a new need for understanding and interpreting the results of such algorithms. In recent years this goal has been worked toward by research in the field of explainable artificial intelligence (XAI) [20].

With this study, we aim to show that high mass resolution MALDI imaging supplemented with deep learning can be used to address the need to differentiate AdCys, which arise in the salivary gland from other types of SGCs. Additionally, we employ methods of explainable AI to identify masses which are significant for the detection of AdCys and match them to possible proteins. The spatially resolved nature of MALDI imaging allows us to compare the mass distributions with other histological methods.

To our knowledge, this approach has for the first time succeeded in performing an automated deep-learning-based classification of AdCys in comparison to other SGC subtypes. Since our analysis is based on MALDI-imaging and therefore includes a spatial component, we were able to analyze our results using methods of XAI and match them with histopathological insights.

## 2. Materials and Methods

### 2.1. TMA Preparation

Nineteen SGC samples (for clinicopathological data see Appendix A) from previously created tissue microarrays (TMAs) were included in the study. Samples were selected and reviewed by two pathologists with special expertise in the field as previously described (PMID: 34378164). Briefly, primary SGC FFPE blocks were retrieved from the archives of the Institute of Pathology at the University Hospital of Cologne. All diagnoses were critically reviewed using a panel of IHC and FISH tests (PMID: 34378164) before inclusion in the study. Four tissue cylinders with a diameter of 1.2 mm per case were included in the TMA. Additionally, we included tonsil and appendix samples from six healthy patients, which served as a technical control (Table 1). Handling of patient material and data were in accordance with the ethical standards of the University of Cologne and the Helsinki Declaration of 1975 and its later amendments. Patients gave their written informed consent to participation.

### 2.2. Sample Preparation

TMA sections (6 µm) were mounted on indium tin oxide (ITO)-coated glass slides. Sample preparation was performed using SunTissuePrep (SunChrom, Friedrichsdorf, Germany). The protocol was consistent with previously published protocols [21]. In brief, the subsequent steps were performed: deparaffinization, washing with xylene and ethanol, pH conditioning, and antigen retrieval. For on-tissue enzyme digestion, a trypsin solution of 2 µg/µL trypsin (Promega Gold) in 50 mM AMBIC buffer and 10% ACN was used. As the spraying device, the SunCollect System (SunChrom, Friedrichsdorf, Germany) was used. Digestion was performed with the SunDigest chamber (SunChrom, Friedrichsdorf, Germany) in basic mode. The matrix used for the experiments consisted of 10 mg/mL CHCA in 60% ACN and 0.2% TFA and was applied with the SunCollect.

### 2.3. MALDI Measurement

A dual source ESI/MALDI-FT-ICR mass spectrometer equipped with a dynamically harmonized analyzer cell (solariX XR, Bruker Daltonics Inc., Billerica, MA, USA) and a 12 T refrigerated actively shielded superconducting magnet (Bruker Biospin, Wissembourg, France) instrument was used for MALDI imaging. The instrument was operated in positive ionization, broadband mode (mass range 590–4000 *m*/*z*) with a 1 MW time domain (FID length: 1.677 s; mass resolution at *m*/*z* 1047: ~175,000). The measurement method was linearly calibrated with the Peptide Mix II (Bruker Daltonics) between 700 and 3200 Da (rms: 0.188 ppm, *n* = 8). The laser settings were optimized for spectral quality and peak magnitude (minimum focus, 9% laser power, 50 shots @ 2 kHz). MALDI imaging data were acquired using FlexImaging version 5.0 and ftmsControl version 2.3.0 (Bruker Daltonics) with a raster size of 50 µm.

### 2.4. Data Preprocessing

Using the SCiLS Lab Software (Bruker Daltonik, Bremen, Germany) we re-binned the individual spectra into equidistant 0.3 mDa bins and a mass range of 590 *m*/*z*–2000 *m*/*z*, which led to 217,984 data points per pixel. We performed no baseline removal or global normalization but normalized the intensities for each pixel to a range between 0 and 1.

With the co-registered microscopy images, we excluded all pixels that were part of a core but contained no tissue.

We also used the SCiLS Lab Software to find the 500 peaks best correlated with the highest pure matrix peak and created a dataset where all values in the range of 0.02 Da of these peaks were set to zero.

### 2.5. Deep-Learning-Based Classification

For the deep-learning task of tumor subtyping we treated each individual spectrum which corresponded to one measured spot as independent measurements with a single label. As described by [22], these individual spectra possess similarities to images and can be processed using many of the same techniques proposed for image classification tasks.

Spectra are also similar to time-series data. Convolutional neural networks excel at solving practical problems on both types of datasets [10].

We evaluated multiple different neural network architectures, including multiple combinations of fully connected and convolutional layers paired with pooling layers and leaky rectified linear units [23] as activation function. All networks were implemented in Pytorch [24] and the training was performed using the Adam optimizer at a learning rate of 0.001. All calculations were performed on a single Nvidia GeForce RTX 2080 Ti.

We evaluated all network architectures using 5-fold -cross-validation with random splits that aimed to preserve the ratio between the eight classes.

### 2.6. Deep Lift

DeepLift is one method of interpreting the results of a deep neural network, which allow for scoring the negative and positive contributions of each input feature to the predictions of a specific label. For a more detailed explanation we refer to the original work [25]. We applied this method to our classification model to identify potential discriminative masses for each tumor subtype. We calculated the 10 input features with the highest positive contribution for each class.

### 2.7. Clustering with densMAP

To visualize the results of the DeepLift analysis, we performed a nonlinear supervised dimensionality reduction using densMAP [26]. This method allowed us to visualize the high dimensional data points in our dataset. It is based on uniform manifold approximation and projection (UMAP) [27] but learns to preserve and takes into account local density in the input space. As features we used the masses calculated in 2.6, and reduced the dimensionality to 2.

### 2.8. Comparison with Microscopy Images and Histological Assessment

Finally, we visualized the distribution of the most significant predicted masses for the class AdCy. Using the SCiLS Lab Software, we overlaid this distribution onto a microscopy image of a core that contained AdCy tissue as well as healthy connective tissue. This allowed us to understand if the algorithm picked tumor-specific features or based its assessment on other characteristics of the sample.

## 3. Results

### 3.1. Network Architecture

On a small subset of the data, we observed the best performance using a convolutional neural net with six convolutional layers, each followed by a max pooling layer, and a final fully connected layer. We used this network architecture for all following experiments.

### 3.2. Classification Results

The final per-class classification accuracy, averaged over all test sets, is presented in Table 2. The highest per-class accuracy was achieved on the control spot, which contained no tissue and was therefore the most homogenous class. The per-class accuracy of all SGC subtypes ranged from 83% to 87%. The lowest accuracy was achieved on the class Anos, which consists of multiple different not-further- specified adenocarcinomas of the salivary gland and shows a high level of heterogeneity.

In Figure 1, we visualize the classification results in the original pixel layout. We show correctly classified spectra represented as pixels in green and mark misclassifications in red. This visually demonstrates that the classification accuracy did not differ between different tumor subtypes or patients but that there were individual cores with higher percentages of misclassified spectra. We did not observe any correlation of the level of correctly classified spectra with histopathological features. We also evaluated the per-core accuracy by choosing the majority label for all pixels in one core. This resulted in 100% correct core-wise classification.

### 3.3. Predicted Masses

We analyzed the contribution of each mass to the prediction of each class using DeepLift. We then focused on the top ten contributions per class and further analyzed our dataset in regard to this list of masses. We filtered the resulting list of 80 masses (10 masses per class with 8 classes, see Appendix A) for duplicates and found that only 64 out of 80 masses appeared only in one tumor entity. The subset that contained duplicates can be matched to a measurement artifact. Therefore, we excluded this set from further analysis.

### 3.4. Clustering with densMAP

In Figure 2 we plot the results of the dimensionality reduction using densMAP. The three SGC entities with the highest number of samples and therefore spectra (Acin, AdCy and MuEp) each show one large cluster containing the majority of samples. All three form additional smaller clusters. Further evaluation shows that these additional clusters all predominantly contain samples from one core. In the Appendix A we provide an interactive version of the above plot in which each point is mapped to its core. For the other SGC subtypes there is less of a clear clustering. This is expected for Anos, since these are a heterogenous group of SGCs. For the Sec, the class with the lowest number of samples, we observed strong inter-core differences. As a result, we get three distinct clusters, one for each Sec core.

### 3.5. Spatial Distribution and Comparison with Microscopic Imaging

We compared the distribution of the mass peak with the strongest positive contribution for the class AdCy to a microscopic view of a core that contained tumor tissue as well as healthy connective tissue (Figure 3). This shows a high relative intensity in the tumor area and allows a differentiation between the two tissue types. With this result we show that the algorithm is picking up tumor-specific features.

## 4. Discussion

In this study we used high-resolution MALDI imaging data from a 12-Tesla instrument for a deep-learning approach. The aim was to provide a high classification accuracy on a challenging dataset consisting of six different SGC tumor subtypes, including AdCy, and healthy control samples.

Until now, high mass resolution MALDI imaging has been used to discriminate between different pathological regions in SGCs [28] but not to differentiate between different subtypes of SGCs. In a similar way, deep-learning approaches to high-resolution MALDI imaging are, to the best of our knowledge, limited to binary classification tasks on mouse models [29], and had not been applied before to tumor subtyping of SGCs.

With the aid of deep learning, we achieved above 80 percent per pixel classification accuracy on all tumor subtypes and a perfect whole-TMA core classification.

To get the most benefit out of the high mass resolution, we further applied techniques of explainable AI. This allowed us to interpret the results and predict masses that are most significant in predicting AdCy and other SGC entities. In this study, we applied DeepLift to get a list of ten masses per class with the strongest positive influence on the correct classification. This provided a better understanding of the results from a black-box-like deep-learning algorithm. Even before using these results for further experiments, this led us to discover a measurement artifact which allowed us to exclude 16 masses from further data analysis.

Using the predicted masses as features for a nonlinear dimensionality reduction method (NLDR) provides an extra layer of interpretability to the otherwise opaque classification approach. NLDR methods such as UMAP are a useful tool to visualize high-dimensional data [30]. We applied densMAP, an augmented version of UMAP which has shown to provide better clustering with regard to the local density of the samples in the original space. The resulting plot shows that the majority of the data points originate from the three most prevalent SGC subtypes in our dataset, forming one large cluster each. A closer inspection reveals that there are single data points of each class appearing in clusters predominantly belonging to another class. This is in line with the high but not perfect classification accuracy of our deep-learning model. By investigating the origin of data points forming the smaller clusters we found that they almost exclusively represented single cores (interactive Appendix A).

In the final step of our analysis, we mapped the mass with the most significant positive contribution to the class AdCy to microscopic images and histopathological annotations. Specifically, we validated our results by analyzing the distribution of this mass on a core that contained AdCy and healthy connective tissue. Since our dataset was composed almost exclusively of tumor tissue, this core presents an edge case. The fact that we observed a distinct distribution only in the tumor tissue of the core validates our approach. Additionally, this allows us to confirm the applicability of our deep-learning algorithm in the presence of small amounts of label noise (we incorrectly labelled the healthy tissue in this core as AdCy). This is consistent with the literature on this topic [31] and supports our core-wise labelling strategy.

While we were able to interpret our classification results and get significant masses for the prediction of different SGC subtypes, we could not reliably match those experimental finds to specific proteins. Further work needs to be done in this protein identification task. Currently available technical and algorithmical solutions and absent dedicated databases do not provide satisfactory and verifiable results in this regard.

In contrast to our approach in which we searched for the presence of masses, another option for further study lies in the identification of masses with significant negative predictive power for certain SGC subtypes.

Regarding the applicability to a clinical setting, a recently published work shows that machine learning algorithms trained on MALDI imaging measurement of TMAs can successfully be transferred to classification tasks on whole sections [32]. While these results were obtained with lung cancer patients, it seems likely that these results hold true for the tissue types and classification tasks presented in our study, since the experimental setup shows strong similarities and in both studies the task is cancer subtyping.

## 5. Conclusions

We were able to show that high mass resolution MALDI imaging in combination with deep learning can be used for tumor subtyping in SGCs. With methods of explainable AI and data visualization strategies based on nonlinear dimensionality reduction, we gained additional insights, for example into the homogeneity of the AdCy proteome among different patients. Additionally we were able to provide an intuitive understanding of deep-learning results by visualization of tumor-specific features combined with histopathological images. These allow us to verify the plausibility of our results and propose specific masses as a basis for further research into marker identification. Given the likely transferability of our TMA-based results, we see the potential for MALDI imaging as an additional diagnostic tool for the demanding task of SGC subtyping on the one hand and a general approach for other cancer entities on the other. To reach this goal, the following points have to be addressed: First, results should be evaluated in a larger independent tumor cohort. Secondly, the “black box” of deep learning can be further opened by methods of explainable AI for a better understanding of carcinogenesis, and in the long run enable better diagnostic and treatment regimes for cancers.

## Figures and Tables

**Figure 1 cancers-14-04342-f001:**
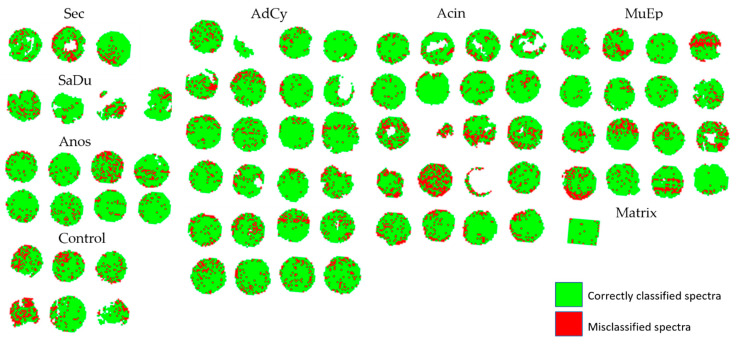
Combined visualization of 5-fold cross-validation showing the results on each test set. Green pixels show correctly classified spectra, red pixels show misclassifications. AdCy: adenoid cystic carcinoma; MuEp: mucoepidermoid carcinoma; SaDu: salivary duct carcinoma; Acin: acinic cell carcinoma; Sec: secretory carcinoma; ANOS: adenocarcinoma not-otherwise-specified.

**Figure 2 cancers-14-04342-f002:**
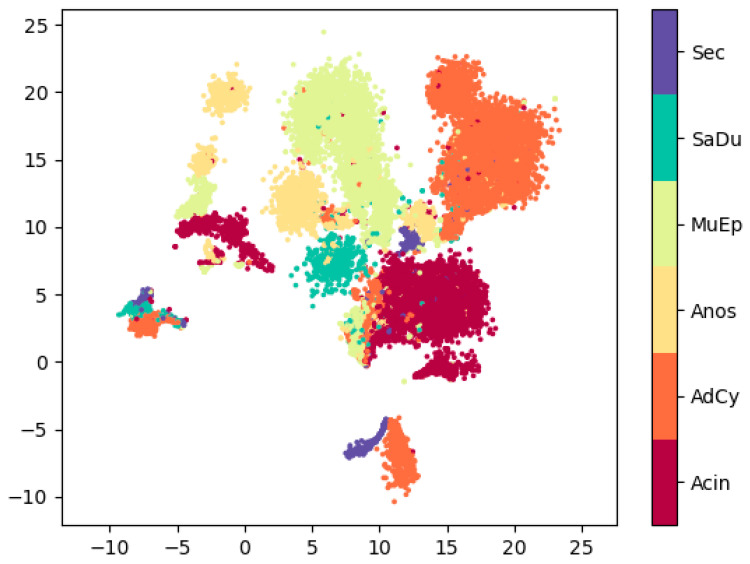
Results of supervised densMAP clustering performed on the dataset. For each pixel the top ten significant peaks per class according to the DeepLift results were used as features. An interactive version of this graph, where each pixel is shown with the core it belongs to, can be found in the supplementary section. AdCy: adenoid cystic carcinoma; MuEp: mucoepidermoid carcinoma; SaDu: salivary duct carcinoma; Acin: acinic cell carcinoma; Sec: secretory carcinoma; ANOS: adenocarcinoma not-otherwise-specified.

**Figure 3 cancers-14-04342-f003:**
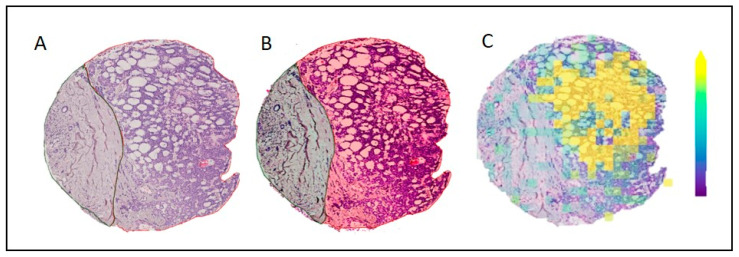
The distribution of a representative mass peak for AdCy in one TMA core and histological annotations. (**A**) Original H&E scan; (**B**) H&E scan overlayed with histopathological annotations (Green: Healthy connective tissue. Red: AdCy tumor tissue); (**C**) Relative intensity of mass 845.47321 *m*/*z* ± 3 mDa.

**Table 1 cancers-14-04342-t001:** Patient and sample overview.

Type	No. Patients	No. Cores
Secretory carcinomas (Sec)	1	3
Salivary duct carcinoma (SaDu)	1	4
Mucoepidermoid carcinoma (MuEp)	4	16
Adenocarcinoma not otherwise specified (Anos)	2	8
Adenoid cystic carcinoma (AdCy)	6	24
Acinic cell carcinomas (Acin)	5	20
Control (human tonsil and appendix)	6	6

**Table 2 cancers-14-04342-t002:** Per-class accuracy of our deep-learning approach averaged on all test sets.

Class	Per Class Accuracy in Percent
Acin	85.288
AdCy	83.96
Anos	83.262
Control	78.782
Matrix	95.476
MuEp	87.032
SaDu	84.35
Sec	87.288

## Data Availability

The original data presented in the study are included in the article. Further inquiries can be directed to the corresponding author.

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
