# Peer review of "Multi-Class Cancer Subtyping in Salivary Gland Carcinomas with MALDI Imaging and Deep Learning"

_cancers, 2022, doi:10.3390/cancers14174342_

Round 1

Reviewer 1 Report

This manuscript is a clear description of the use of high performance MALDI-MS for tumor classification using the AI techniques described.  The tumor classification results are at the 80% level, which may or may not be better than other classification methods which are claimed to be inadequate.  I am not familiar with the accuracy of those, so will not comment on such a comparison. Overall, I believe that publication of this study is adequately justified.

Author Response

Reviewer 1

This manuscript is a clear description of the use of high performance MALDI-MS for tumor classification using the AI techniques described.  The tumor classification results are at the 80% level, which may or may not be better than other classification methods which are claimed to be inadequate.  I am not familiar with the accuracy of those, so will not comment on such a comparison. Overall, I believe that publication of this study is adequately justified.

Response: We thank the reviewer for the encouraging comments and for reviewing our manuscript. The classification results of about 80% are in general not better than other classification methods, but to the best of our knowledge, they are the best using an automated, deep learning based classification to separate AdCys from other SGC subtypes. We clarified this in the introduction (Line 76-79).

Reviewer 2 Report

In this study the authors use MALDI imaging together with deep learning to classify subtypes of salivary gland tumors, a heterogenous group of entities for which diagnosis can be challenging. Their model shows excellent per-class accuracy but includes very few patients of each subtype and is not evaluated in a larger independent tumor cohort. They further fail to assign which proteins are responsible for the class separation, information that would be very useful to evaluate their findings as well as have potential use in conventional histopathological classification. The study lacks clinicopathological descriptions of the patients. It is further not clear why the authors use tissue from tonsil and appendix as their control tissue. Undoubtedly, normal salivary gland tissue should be used in the model. Notably, it was not possible to access and evaluate Figure S1 online. In my opinion this study is premature and needs further work.

Author Response

Reviewer 2

In this study the authors use MALDI imaging together with deep learning to classify subtypes of salivary gland tumors, a heterogenous group of entities for which diagnosis can be challenging.

General Response: We thank the reviewer for reviewing our manuscript and for the constructive comments. These comments have helped us to improve the quality of the manuscript.

Point 1: Their model shows excellent per-class accuracy but includes very few patients of each subtype and is not evaluated in a larger independent tumor cohort.

Response 1: Salivary gland carcinomas (SGC) are rare and therefore would take years to evaluate large samples of each SGC subtype. Instead, we showed that even a smaller sample oft the used SGC samples, which were very well characterized in former studies, built a solid and statistically robust base for MALDI imaging using a 12 Tesla magnetic resonance mass spectrometer in combination with subsequent deep learning classification.

Point 2: They further fail to assign which proteins are responsible for the class separation, information that would be very useful to evaluate their findings as well as have potential use in conventional histopathological classification.

Response 2: We fully agree with this assessment. Nevertheless, there is no 100% reliable way to perform protein identification based on MALDI-Imaging spectra. Software solutions like MS-Fit or Mascot Server offer several putative identifications. Such identifications go far beyond the aim of the present study. To allow other groups to work towards protein identification we added a table (S2) to the Supplementary Materials, presenting the most significant masses for each tumor subclass (as discussed in 3.3).

Point 3: The study lacks clinicopathological descriptions of the patients.

Response 3: We thank the reviewer for this remark and added the clinicopathological details (S1) in the Supplementary Materials.

Point 4: It is further not clear why the authors use tissue from tonsil and appendix as their control tissue. Undoubtedly, normal salivary gland tissue should be used in the model.

Response 4: We have clarified in the introduction (Line 90 - 92), that the included tonsil and appendix tissue purely serves as a technical control and is not used in the later analysis. Normal salivary gland tissue was not essential, because we wanted to establish an accurate tumor subtyping, not the differentiation between tumor and normal tissue.

Point 5: Notably, it was not possible to access and evaluate Figure S1 online.

Response 5: Thank you for pointing this out to us. This interactive figure S3 (formerly S1) is important for understanding this study. It allows the corresponding localizations to be displayed on the TMA for the individual data points in Figure 2. We are able to reproduce the problem if the figure is downloaded from the MDPI server. We are trying to solve the problem with the editor and the production team.

Point 6: In my opinion this study is premature and needs further work.

Response 6: We take the criticism seriously, but we are of the opinion that at least two facts justify publication: 

(1) This approach has succeeded for the first time to perform an automated, deep learning-based classification of AdCys in comparison to other SGC sub-types with high accuracy.

(2) Since MALDI-Imaging is a measurement modality with a spatial component, we could use methods of explainable AI (XAI) to match our results with histopathological annotations. This is illustrated in figure 3.

Reviewer 3 Report

In this work authors talks about MALDI Imaging and Deep Learning for Salivary Gland Carcinomas. I found the methodological part to be well justified and reasonable for this type of analysis. Although the manuscript is overall well-written and structured, it might benefit from additional spell/language checking.

Title can be improved.

The introduction is deprived of the related work with the recent literature.

There are several interesting papers that look into Deep learning in healthcare. For instance, the below papers has some interesting implications that you could discuss in your Introduction and how it relates to your work.

Vulli, A.; et al.. Fine-Tuned DenseNet-169 for Breast Cancer Metastasis Prediction Using FastAI and 1-Cycle Policy. Sensors 2022, 22, 2988.

Ali, Farman, et al. "A fuzzy ontology and SVM–based Web content classification system." IEEE Access 5 (2017): 25781-25797.

Srinivasu, Parvathaneni Naga, et al. "Classification of skin disease using deep learning neural networks with MobileNet V2 and LSTM." Sensors 21.8 (2021): 2852.

Authors should further clarify and elaborate novelty in their contribution. Best is to put them in 2nd last paragraph of the introduction.

What are the key issues present study has addressed?

Authors talks about deep learning, but specifically mention which algorithm they have used.

Do mention what is the pseudo code. Why authors choose particular approach and why not others? Provide references to support your claims.

Lines 267-268, "It seems likely that these results hold true for the tissue types and classification tasks present in our study"... This statement is very open ended, kindly explain in detail with references.

What are the practical implications of your research?

Conclusion is too short. Add more explanation.

What are the limitations of the present work?

Author Response

Reviewer 3

In this work authors talks about MALDI Imaging and Deep Learning for Salivary Gland Carcinomas. I found the methodological part to be well justified and reasonable for this type of analysis. Although the manuscript is overall well-written and structured, it might benefit from additional spell/language checking.

General Response: We thank the reviewer for reviewing our manuscript and for the constructive comments. These comments have helped us to improve the quality of the manuscript.

Point 1: Title can be improved.

Response 1: We discussed internally the title again. We believe that the title “Multi-Class Cancer subtyping in Salivary Gland Carcinomas with MALDI Imaging and Deep Learning” contains all for the reader important information. We are, nevertheless, open to a suggestion of the reviewer.

Point 2: The introduction is deprived of the related work with the recent literature.

Point 3: There are several interesting papers that look into Deep learning in healthcare. For instance, the below papers has some interesting implications that you could discuss in your Introduction and how it relates to your work.

Response 2+3: We thank the reviewer for raising this issue. We have added text in the introduction to address this point and have included three additional related works with recent literature in the introduction as suggested. Especially the first mentioned paper (Vulli et al; citation 18) is a highly interesting recent development to a topic that we already addressed in the introduction (Line 63).

Point 4: Authors should further clarify and elaborate novelty in their contribution. Best is to put them in 2nd last paragraph of the introduction. What are the key issues present study has addressed?

Response 4: We have now better clarified and elaborated the novelty of our contribution and highlighted which key issues this study addresses (Line 75 - 79).

Point 5: Authors talks about deep learning, but specifically mention which algorithm they have used.

Response 5: We describe the specifics of our deep learning implementation in section 2.5 in detail.

Point 6: Do mention what is the pseudo code. Why authors choose particular approach and why not others? Provide references to support your claims.

Response 6: We thank the reviewer for this comment. Since we use already established machine learning methods, we do not focus on the algorithmical details. In the introduction we have added additional literature to support our choice of convolutional neural networks (CNN) for our classification task and provide further references to the applicability of CNNs to the spectral data obtained in MALDI-Imaging in 2.5.

Point 7: Lines 267-268, "It seems likely that these results hold true for the tissue types and classification tasks present in our study"... This statement is very open ended, kindly explain in detail with references.

Response 7: We have improved this statement by specifying in which ways the quoted study resembles ours, and where the differences lie (Line 277 - 280).

Point 8: What are the practical implications of your research?

Response 8: We see the main practical implications of our research in the potential to combine deep learning with MALDI imaging as an additional diagnostic tool for the demanding task of SGC subtyping on the one hand and a general approach for other cancer entities on the other hand. We now describe this in more detail in the conclusions.

Point 9: Conclusion is too short. Add more explanation. What are the limitations of the present work?

Response 9: We thank the reviewer for his input and have added additional details to the conclusion, including a discussion of the limitations of our work, as well as the methods used in general.

Round 2

Reviewer 2 Report

The authors did not address the major concerns regarding validation of the model in an independent cohort or presenting novel useful biomarkers for the tumors.

Reviewer 3 Report

All my comments are addressed. Hence, paper is acceptable now.